# PPARδ Agonist Promotes Type II Cartilage Formation in a Rabbit Osteochondral Defect Model

**DOI:** 10.3390/cells11192934

**Published:** 2022-09-20

**Authors:** Ju-Yong Song, Jae-Suh Park, Joo-Hwan Kim, Joon-Ho Wang, Holly C. Heck, Bruce E. Heck, Dong-Hyun Kim, Keon-Hee Yoo

**Affiliations:** 1Department of Health Sciences and Technology, SAIHST, Sungkyunkwan University, Seoul 06355, Korea; 2Department of Pediatric Hematology/Oncology, Samsung Medical Center, School of Medicine, Sungkyunkwan University, Seoul 06351, Korea; 3Department of Orthopedic Surgery, Samsung Medical Center, School of Medicine, Sungkyunkwan University, Seoul 06351, Korea; 4Northwest Ohio Orthopedics and Sports Medicine, Findlay, OH 45840, USA; 5Department of Orthopedics, College of Natural Science, University of Toledo, Toledo, OH 43614, USA

**Keywords:** osteoarthritis, type II hyaline cartilage, type II collagen, peroxisome proliferator-activated receptor delta, 3D collagen scaffold, mesenchymal stem cells

## Abstract

Osteoarthritis (OA) is a chronic degenerative joint disease accompanied by an inflammatory milieu that results in painful joints. The pathogenesis of OA is multifactorial, with genetic predisposition, environmental factors, and traumatic injury resulting in the direct or indirect loss of cartilage. The articular cartilage can also be damaged by direct focal traumatic injury. Articular cartilage provides a smooth, deformable bearing surface with a low coefficient of friction, increased contact area, and reduced contact stress. Articular type II hyaline cartilage lines the synovial joints and, when injured, has a limited ability for repair, except for the most superficial layers via diffusion from the synovial fluid, secondary to no blood supply, a complex structure, and a low metabolic rate. Restoring the articular surface can relieve pain and restore function. Although many strategies have been developed to regenerate type II collagen based on the extent of the lesion, surgical treatments are still evolving. The peroxisome proliferator-activated receptor delta (PPARδ) agonist and collagen treatment of mesenchymal stem cells (MSCs) enhance the chondrogenic capacity in vitro. We present a novel technique for cartilage restoration in a rabbit cartilage osteochondral defect model using a PPARδ agonist (GW0742)-infused 3D collagen scaffold to induce type II cartilage from MSCs.

## 1. Introduction

Peroxisome proliferator-activated receptors (PPARs) are nuclear hormone receptors that form heterodimers with retinoic X receptors to activate or suppress downstream target genes depending on ligands and coactivators. There are three isoforms currently known: PPARα (NR1C1), PPARβ/δ (NR1C2), and PPARγ (NR1C3) [1]. PPARs exert profound effects secondary to their gene regulation and have captured the attention of researchers and clinicians for years. PPAR alpha and gamma agonists have been used for the clinical treatment of dyslipidemia and diabetes [2], whereas PPARβ/δ agonists have been controversial due to concerns about their pro-inflammatory and pro-tumorigenesis effects in vitro [3]. Moreover, they have not been utilized clinically as a therapeutic target for metabolic syndrome or osteoporosis, which would require systemic administration [3]. However, there are no in vivo models supporting PPARβ/δ agonist tumorigenesis [4].

We have demonstrated the beneficial effects of a PPARδ agonist on mesenchymal stem cells (MSCs) by inducing the production of type II collagen-producing chondrocytes in human arthritic synovial fluid [5]. We have also synthesized a directed library of compounds with structure–activity relationships of PPARδ activation relative to human MSC osteogenesis and in a mouse model of human osteoporosis that included toxicity assessments of PPARδ agonists, including GW0742, at the biochemical level, in an in vitro efficacy-related cell culture, and in an in vivo model, in which none of the compounds demonstrated signs of gross toxicity [6]. In addition, an independent toxicity examination of several of our PPARδ compounds, including GW0742, supported the conclusion that GW0742 demonstrates “cytotoxic potential” at 100 µM, and the lower threshold for potential toxicity for some PPARδ agonists is approximately 50 times higher than its efficacious dose range in an MSC assay [5]. Because PPARδ toxicity appears to be dose dependent and PPARβ/δ is ubiquitously expressed throughout the tissues, with higher expression in musculoskeletal tissue and few species differences [7], PPARβ/δ administration for musculoskeletal applications, such as the promotion of osteogenesis for fracture healing and the induction of MSCs to chondrocytes to promote cartilage regeneration, is appealing [3].

Three-dimensional (3D) printing technology has revolutionized many areas of the manufacturing world and is currently being applied in the medical field. The advantage of 3D printing is that a product can be printed using biomaterials such as collagen, hydrogel, fibrinogen, fibrin, or poly(ethylene glycol) dimethacrylate (PEGDMA) and can be customized depending on the size or shape of the patient’s needs or, more practically for the patient and surgeon, can be prefabricated with various shapes or sizes, such as discs or sponges of various diameters and thicknesses, that could be selected and implanted at the time of surgery when necessary [8]. An additional advantage of 3D printing is the ability to add small molecules to the 3D scaffold that signal the host’s MSCs to differentiate into bone or cartilage [9]. A technology that could be implemented at the time of an index procedure and easily used by all surgeons skilled in their trade to reproduce type II collagen after a post-traumatic injury or osteochondral defect, without having to harvest cartilage from a donor site, utilize allograft with the risk of infection and rejection histocompatibility, or having to expose the patient to a second operation such as autologous cultured chondrocytes on a porcine collagen membrane or matrix-induced autologous chondrocyte implantation (MACI), would be a significant discovery [10]. 

Osteoarthritis (OA) is a major contributor to disability in the United States and the world, with an expected dramatic increase over the next 20 years due to an increasingly aging population [11]. Consequently, much attention has been focused on enhancing the chondrogenic capacity of MSCs to regenerate the cartilage of OA patients. Previous studies have revealed that the treatment of MSCs with a PPARδ agonist enhances chondrogenic capacity in vitro [12]. It is known that co-cultures of MSCs with extracellular matrix components, such as hyaluronic acid (HA) or collagen, enhance the chondrogenic capacity of MSCs [13,14]. We hypothesized that combining the PPARδ agonist, GW0742, with a 3D collagen scaffold would enhance chondrogenic differentiation. Based on these previous studies and our knowledge of 3D printing, PPARδ-infused 3D collagen scaffolds were used to promote type II collagen formation in a rabbit osteochondral defect model. 

## 2. Materials and Methods

### 2.1. Reagents

The PPARδ agonist, GW0742 (Cat. DC4292), was purchased from Biobyt (Cambridge, UK), TGF-β was purchased from R&D Systems (Minneapolis, MN, USA), and collagen solution (Cat. #04902) was purchased from STEMCELL Technologies, Inc. (Vancouver, BC, Canada). Minimum essential medium (MEM)-α, Dulbecco’s modified Eagle medium (DMEM), fetal bovine serum (FBS), antibiotic–antimycotic solution, and trypsin–EDTA were purchased from Gibco (Grand Island, NY, USA), and phosphate-buffered saline (PBS) was purchased from Biowest (Riverside, MO, USA).

### 2.2. Culture of Wharton’s Jelly-Derived MSCs

In this study, human Wharton’s jelly-derived MSCs (WJ-MSCs) were obtained from the Samsung Stem Cells and Regenerative Medicine Institute (Seoul, Korea). At 70% confluency, cells were washed twice with PBS, resuspended in 0.05% trypsin–EDTA, and re-plated into a new flask at a density of 3000–5000 cells/cm^2^ in MEM-α (Gibco, Grand Island, NY, USA) supplemented with 10% FBS and 1% antibiotic–antimycotic solution.

### 2.3. Bio-Printing Scaffold (With MSCs)

MSCs were suspended in DMEM, and 6% collagen solution was prepared in an independent tube. Subsequently, the cell suspension and collagen solution were mixed in a 1:1 ratio to print the scaffold using Dr. Invivo 4D2 (Rokit Healthcare Inc., Seoul, Korea) by following the manufacturer’s instructions. Finally, 5 × 10^5^ MSCs were embedded into each scaffold. Different concentrations of GW0742 (a PPARδ agonist) or TGF-β were dissolved in DMEM containing MSCs and embedded in the scaffold in the experimental group. The scaffold was manipulated to be 5 mm in diameter and 3 mm in depth for the experiments.

### 2.4. Chondrogenesis of Printed Scaffold Culture

Groups of WJ-MSCs in collagen scaffolds were incubated in chondrogenic medium (Gibco, Stempro, Grand Island, NY, USA) with or without different concentrations of the PPARδ agonist, GW0742, or TGF-β for 14 days. The medium was changed every 3 days. Normal rabbit cartilage was used as a control to compare the chondrogenesis of WJ-MSCs in the collagen-based scaffold. The scaffold and rabbit cartilage were fixed with 10% formalin solution and stained with 1% alcian blue to analyze chondrogenesis. 

### 2.5. Reverse Transcription-Polymerase Chain Reaction (RT-PCR)

The total RNA was isolated using a RNeasy mini kit (Qiagen, Valencia, CA, USA) according to the manufacturer’s instructions. Complementary DNA (cDNA) was synthesized with 5 µg of RNA using an AccPower RT PreMix kit (Bioneer, Daejeon, Korea) according to the manufacturer’s instructions. 

The resulting cDNA was amplified by PCR with AccuPower^®^ PCR PreMix (Bioneer, Daejeon, Korea). The PCR primers and conditions are summarized in Table 1. The PCR products were analyzed by 1.5% agarose gel electrophoresis and the amplification signal from the target gene was normalized by the glyceraldehyde-3-phosphate dehydrogenase (GAPDH) signal in the same reaction.

### 2.6. Animal Experiments

This study was approved by the Institutional Animal Care and Use Committee of the Samsung Medical Center, South Korea (IACUC No. 20190529002). Male New Zealand white rabbits (Jackson Laboratories, Bar Harbor, ME, USA), 8–9 weeks old and 2.5–3.0 kg, were used in this study. To minimize distress, the rabbits were housed for at least seven days. Animals were randomly assigned to five groups and anesthetized with ketamine (60 mg/kg) and xylazine (5 mg/kg). Cartilage defects were surgically induced at the trochlear groove and the distal femur. The defect size was 5 mm in diameter and 3 mm in depth, identical to the size of the scaffold. 

### 2.7. Histological Analysis and Scoring Cartilage

After 12 weeks of bio-scaffold implantation, all rabbits were sacrificed, and the calvarial defects were carefully dissected and harvested from each animal. The specimens were fixed with 10% formalin solution for two days and decalcified in decalcification acid reagent (Thermo Fisher Scientific, Kalamazoo, MI, USA). The samples were then dehydrated in a graded series of ethanol solutions and embedded in paraffin. Each sample was serially sectioned (4 μm) along the midline of the calvarial defects and stained with hematoxylin and eosin (H&E) or type II collagen (COL2) to evaluate cartilage recovery or collagen content.

A macroscopic and histological evaluation of the cartilage recovery was performed by three independent individuals according to the International Cartilage Repair Society (ICRS) grading system, and the recovery of cartilage was evaluated in four categories: degree of defect repair, integration into the border zone, macroscopic appearance, and overall repair assessment [15]. 

### 2.8. Statistical Analysis

Data are expressed as means ± standard deviation (SD). All data were statistically analyzed using one-way analysis of variance (ANOVA). All experiments were independently performed in triplicate. Statistical analyses were performed using GraphPad Prism, version 6.0. In all groups, *p*-values < 0.05 indicated statistical significance. 

## 3. Results

### 3.1. Synthesis of Bio-Printed WJ-MSCs Scaffold

Previous studies have revealed that collagen-based hydrogels enhance the chondrogenesis of MSCs in vivo and in vitro. We evaluated whether the PPARδ agonist GW0742 could induce the chondrogenesis of WJ-MSCs in 3D conditions. We synthesized a WJ-MSC-based bio-scaffold using 3D printing technologies (Figure 1a). WJ-MSCs were suspended at a density of 3 × 10^6^ cells/mL in DMEM without antibiotics or serum, and a 6% collagen solution was prepared in separate tubes. Each tube was loaded into a bio-printer (DR. INVIVO 4D2, Rokit Inc., Seoul, Korea), mixed in a 1:1 ratio, and printed as a cylinder-shaped scaffold with a diameter of 5 mm and depth of 3 mm. Each scaffold contained 5 × 10^5^ WJ-MSCs. To evaluate whether cells in the scaffold were intact and possessed a capacity for chondrogenic differentiation, we examined the macroscopic and microscopic observations until day 15 of inducing chondrogenesis. The macroscopic observation of the printed scaffold indicated a clear and porous surface (Figure 1b). Microscopic observation of the scaffold indicated that MSCs were present on days 0 and 15. Moreover, the chondrogenic differentiation of the scaffolds was evaluated by alcian blue staining at the end of the experiment. The microscopic observations and chondrogenic differentiation revealed that aggregated MSCs were present in the scaffold without any morphological change until day 15 (Figure 1b,c). The final product of the bio-printed scaffold exhibited a smooth surface and was positive for alcian blue staining (Figure 1d). These results indicated that the scaffold did not affect the morphology and chondrogenic capacity of MSCs. To perform the experiments, scaffolds with different concentrations of chondrogenesis enhancers (TGF-βs) or the PPARδ agonist GW0742 were prepared using MSCs [16,17].

### 3.2. PPARδ Agonist GW0742 Increases Chondrogenesis of Printed Scaffold WJ-MSCs

Previous studies [3] have demonstrated that in vitro cultures of WJ-MSCs with the PPARδ agonist GW0742, alone or with HA-induced chondrogenesis, more potently compared with the TGF-β-treatment. However, it is unclear whether GW0742 can improve the chondrogenic capacity when embedded in scaffolds.

To evaluate the effect of GW0742 on the chondrogenesis of the bio-printed scaffold, we compared it with a TGF-β-containing scaffold. 

We generated WJ-MSC scaffolds in the absence or presence of GW0742 or TGF-β at different concentrations (Figure 2a). After three weeks of chondrogenesis, all groups were stained with alcian blue to evaluate the chondrogenesis of the scaffolds. 

The results indicated that all groups were positive for alcian blue staining, and the bio-printing of WJ-MSCs in collagen did not impair the capacity of chondrogenesis, even though they were embedded in the scaffold.

Scaffolds with TGF-β and GW0742 showed the strongest staining of alcian blue, similar to rabbit cartilage alongside the WJ-MSC-only group (Figure 2b,c).

Among the various concentrations of chondrogenesis enhancers, all scaffolds with GW0742 and 10 ng of TGF-β showed the strongest staining compared to the other groups. This result indicates that the PPARδ agonist GW0742 can potently elevate the chondrogenesis of bio-printed scaffolds more than any scaffold with TGF-β. Moreover, the expression of type II collagen in each scaffold indicated that the scaffold with TGF-β or GW0742 potently induced the expression of type II collagen in the differentiated scaffold (Figure 2d).

### 3.3. Implantation of Bio-Printed Scaffold Enhances Cartilage Recovery

To verify whether the results of the in vitro model can be reproduced in a rabbit model, we established a cartilage defect model using a surgical procedure at the trochlear groove and distal femur. The scaffold was then implanted into the defect. The concentration of the scaffold with TGF-β was 10 ng/mL and that of the scaffold with GW0742 was 1 μM/mL (Figure 3a).

After 12 weeks of implantation, the animals were sacrificed and the cartilage was harvested to evaluate the regeneration by the scaffolds. Gross observations of the cartilage revealed that the surface of the defect-only group was not fully filled, and the border of the defects was still distinct compared to the other groups. In contrast, the implantation of scaffolds enhanced the filling of the defect site (Figure 3b), which is consistent with scaffolds providing a suitable environment for cartilage tissue regeneration [15].

However, the scaffolds with TGF-β or GW0742 showed a clear surface, and the margin borders in the TGF-β or GW0742-containing scaffold groups were well-integrated with normal tissue. 

Similarly, the mean gross International Cartilage Repair Society (ICRS) appearance score for cartilage recovery by scaffold implantation indicated that the scaffolds with TGF-β or GW0742 had higher scores than the other scaffolds. In addition, the scaffold with GW0742 showed the highest ICRS score (Figure 3c) [17]. 

This result indicated that the treatment of the scaffold with GW0742 enhanced cartilage recovery in an animal model.

### 3.4. Quantitative Evaluation of Histological Analysis

The macroscopic regeneration of cartilage was correlated with the structural regeneration of cartilage, which was analyzed by H&E staining (Figure 4a).

The defect-only group showed wound closure by fibrotic tissue with distinct tissue organization from the normal tissue and scaffold groups. In contrast, groups with any scaffold showed organized tissue filling with diverse cells compared to that of the defect-only group. In detail, these groups showed cartilage filled with hyaline-like cartilage morphology in the superficial layer and immature or mature chondrocytes in the deeper layer.

Furthermore, the scaffold with GW0742 revealed that the height of the superficial tangential zone in the damaged tissue regenerated similar to normal cartilage, and the content of hyaline-like cartilage increased. In contrast, the histological analysis of tissue from the scaffold-only group showed immature chondrocytes, and the group with the scaffolds with TGF-β showed columnar-like chondrocytes. 

The ICRS scoring of stained tissues showed that the scaffold with GW0742 reached the highest score.

These results indicated that the scaffolds enhanced the regeneration of the cartilage differently, and that the regeneration by the scaffolds with GW0742 displayed mechanical characteristics similar to those of a normal (or undamaged/uninjured) site.

### 3.5. PPARδ Bio-Scaffold Induces Regeneration of Cartilage by Enhancing the Accumulation of Type II Collagen

In the complex structure of cartilage, the formation of the extracellular matrix plays a key role in normal cartilage function. Since type II collagen (Col2) expression on the articular surface and superficial zone of cartilage is responsible for tensile strength and resistance to shear forces, the formation of Col2 in cartilage is a crucial indicator of regeneration. We evaluated and compared the Col2 expression in the tissues (Figure 5a,b). The results implied that the collagen staining of the recovered cartilage surface in the defect or scaffold-only groups was discontinuous or incomplete. In contrast, scaffolds with both TGF-β and GW0742 showed a continuous layer of cartilage surface and collagen-positive cells in the deep layer. Consequently, these results indicated that GW0742 was more potent than TGF-β at inducing cartilage regeneration.

## 4. Discussion

This study evaluated the effect of a PPARδ agonist with a bio-printed MSCs scaffold for cartilage regeneration. The results revealed that PPARδ agonists could be used to induce type II hyaline cartilage regeneration in an animal model. 

Type II cartilage regeneration that demonstrates normal mechanical properties following an injury has long been sought by surgeons treating these injuries, as well as their patients [18]. Unfortunately, it remains elusive due to the limited ability of cartilage to repair itself secondary to the limited blood supply and the pro-inflammatory microenvironment. 

Mechanical damage and inflammation result in increased levels of pro-inflammatory cytokines in the joint and surrounding cartilage following an initial injury [19], such as interleukin (IL)-1β, IL-6, IL-8, and tumor necrosis factor (TNF)-α. This pro-inflammatory microenvironment following an acute joint injury, as well as chronic conditions such as OA, affects the ability of MSCs to respond favorably and heal injuries, resulting in abnormal differentiation and causing fibroblast-like cells to form fibrocartilage with inferior mechanical properties [20].

We have demonstrated the beneficial effects of a PPARδ agonist on MSCs by inducing the production of type II collagen-producing chondrocytes in human arthritic synovial fluid, and that manipulation of the microenvironment through small-molecule cell signaling is a key factor in the regeneration of normally functioning type II hyaline cartilage [3].

The current MSC-based post-traumatic OA is limited by a strong chondrogenic inducer and a stable scaffold [3]. We have demonstrated that the PPARδ agonist has strong chondrogenic properties, which provides a follow-up to our previous work and hypothesis that scaffolds and matrices with growth factor adjuvants may prove critical for the appropriate localization of MSCs and for stemming the inflammatory response to initiate a regenerative response [21].

There needs to be a balance between sufficient initial inflammation to start the healing process followed by the tight regulation of resident cells, such as MSCs, chondrogenic cells, fibroblasts, and immune cells, in the damaged tissue to regenerate into normal tissue [22] and to avoid a pro-inflammatory degenerative state.

Multiple attempts have been made to regenerate damaged cartilage tissues with the application of MSCs to the area of cartilage damage and synovial fluid; however, the use of MSCs alone to regenerate type II hyaline cartilage has been unsuccessful [18]. 

There are many reasons why stem cells alone have failed, including different degrees of regeneration because of the heterogeneity of cells, donor variation, the tissue of origin, single cells, clones, and the differentiation process [23].

Although MSCs play an essential role in immune regulation, tissue regeneration, and differentiation, they are also affected by their surrounding environment. Thus, controlling the regenerative healing microenvironment through cell signaling and scaffolds in the otherwise hostile host environment of a damaged pro-inflammatory joint is of paramount importance.

This study aimed to evaluate the regenerative capacity of cartilage after implanting a mesenchymal stem-cell-embedded scaffold with the PPARδ agonist GW0742 in a rabbit model. The results revealed that the presence of PPARδ in the scaffold significantly increased the degree of regeneration compared to that of the scaffold with TGF-β. This is consistent with our previous findings that PPARδ agonists are strong chondrogenic inducers of MSCs in the inflammatory OA microenvironment [1]. 

However, we could not evaluate the expression level of regenerative cytokines such as TGF-β1, TGF-β3, IGF-1, and FGF-2 in the animal model [24]. This is because the implantation of a scaffold does not induce the proteins or growth factors that induce regeneration of the damaged tissue, but MSCs in the scaffold itself differentiate into chondrocytes in the damaged site, and the PPAR agonist GW0742 boosts this differentiation. This leads to more local control of the damaged environment by supplying a strong chondrogenic inducer and scaffold with MSCs to restore the delicate balance between the pro-inflammatory state, which has been well-documented in the literature, following cartilage damage/OA and the regenerative process [19,20].

A previous study revealed that the PPARδ agonist GW0742 enhances the chondrogenic capacity by inhibiting PPARγ and inducing the expression of TGF-β and type II collagen in MSCs [1,12]. 

The use of PPARβ/δ agonists has been controversial due to concerns about pro-inflammatory and pro-tumorigenesis effects in vitro. However, PPARδ toxicity appears to be dose-dependent [7] and there are known variations between similar drugs within a certain class, such as telmisartan, an angiotensin II receptor blocker that is a PPARγ partial agonist [25]. It is likely that not all PPARδ are carcinogenic, and additional research is needed to clarify which drugs in this class can be safely used.

Although PPARβ/δ agonists may not be beneficial for clinical use to treat type II diabetes or dyslipidemia, considering the risks when administered systemically [26], PPARβ/δ administration for musculoskeletal applications, such as the promotion of osteogenesis for fracture healing and the induction of MSC differentiation to chondrocytes to promote cartilage regeneration, is appealing. In addition, consideration should be given to the fact that the effect of PPARδ agonists on MSCs in this study differs from the concerns mentioned above for several reasons. First, the MSCs and PPARδ agonist GW0742 were contained in the collagen-based scaffold and did not produce systemic effects. Second, the shape and size of the scaffold were precisely tailored to fit the damaged cartilage and implanted into the damaged site, providing a secure, stable construct for integration into the host tissue and local control of MSC differentiation. Third, the cartilage microenvironment influenced the differentiation of MSCs into chondrocytes, which possess a low proliferation rate and metabolic activity. Lastly, PPARδ agonists possess immunomodulatory properties, such as anti-inflammatory properties, which reduce the activation of infiltrated macrophages [27] and restore balance to the pro-inflammatory state seen in cartilage damage models so that the type II collagen can develop. The use of PPARδ with MSC-embedded scaffolds in type II hyaline cartilage regeneration has been demonstrated as a promising method in a rabbit model without side effects.

Future studies should evaluate the biomechanical properties of the regenerated cartilage and compare the efficacy of other PPARδ agonists that possess similar properties, with the hope of achieving type II hyaline cartilage regeneration with native biomechanical properties for human use. 

## Figures and Tables

**Figure 1 cells-11-02934-f001:**
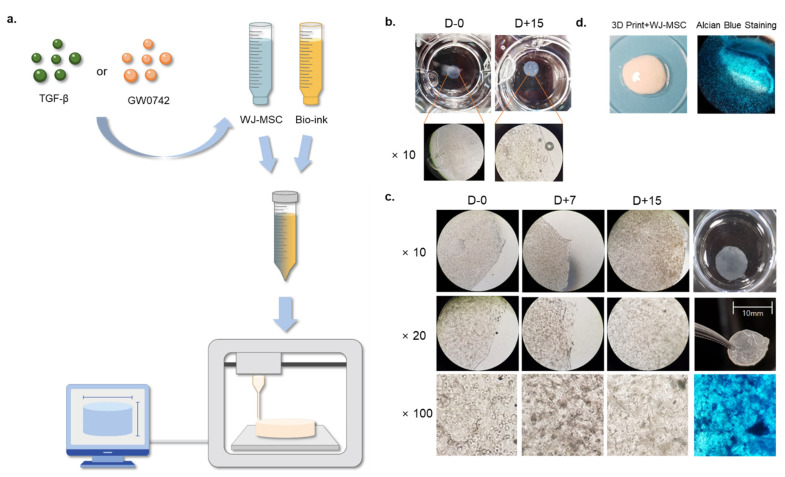
Generation and observation of cartilage-implantable bio-printed scaffolds. (**a**) Schematic showing a generation of the MSC scaffold with the presence or absence of TGF-β or GW0742; (**b**) representative image of macroscopic and microscopic observations of bio-printed scaffolds on day 0 and day 15; (**c**) representative image of microscopic observations of bio-printed scaffolds on days 0, 7, and 15, with alcian blue staining of chondrogenic differentiation on day 15; and (**d**) representative image of bio-printed scaffold and alcian blue staining after chondrogenesis.

**Figure 2 cells-11-02934-f002:**
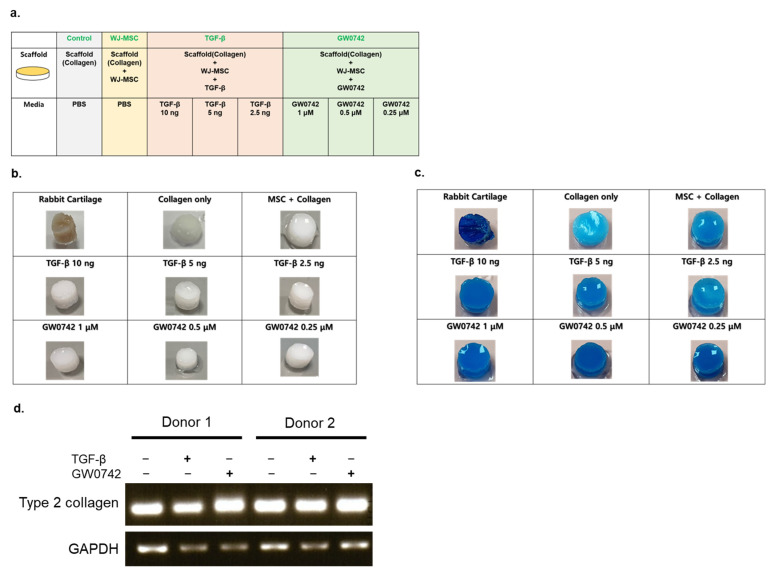
Ex vitro evaluation of chondrogenic differentiation capacity. (**a**) Illustrative scheme describing the experimental design; (**b**) macroscopic observation of collagen-based scaffolds seeded with WJ-MSCs with the presence or absence of TGF-β or GW0742; (**c**) alcian blue staining of the scaffolds cultured with chondrogenic differentiation medium for 28 days; (**d**) reverse-transcription PCR analysis of type II collagen expression from MSCs derived from 2 different donors in the scaffold presence or absence of chondrogenesis inducers (10 ng/mL of TGF-β or 1 μM/mL of GW0742).

**Figure 3 cells-11-02934-f003:**
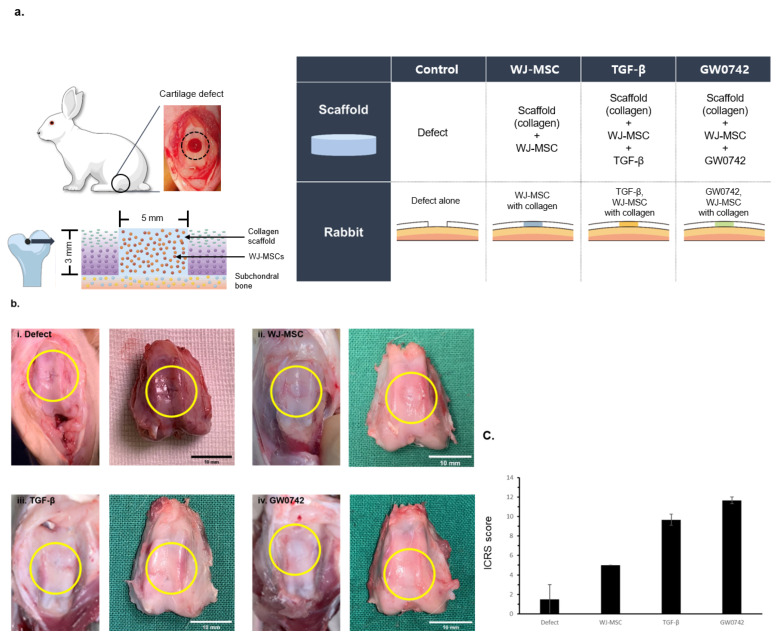
Gross observations of cartilage defect and regeneration by bio-printed scaffolds. (**a**) Scheme of cartilage defect generation and constitution of experimental group. Eight- to nine-week-old New Zealand white rabbits were divided into four groups (defect, scaffold, scaffold with TGF-β, and scaffold with GW0742). Cartilage defects with a 5 mm diameter and 3 mm depth at the trochlear groove and distal femur were generated by surgery. The same size of bio-printed scaffold was implanted at the defect. At 12 weeks of implantation, rabbits were sacrificed to analyze cartilage regeneration. (**b**) Macroscopic observations of repaired defects in the four groups at 12 weeks after implantation (i. defect, ii. WJ-MSCs, iii. TGF-β, iv. GW0742; the yellow circle indicates the site of the defect); scale bar: 10 mm. (**c**) Macroscopic ICRS scores and comparison of each group at 12 weeks after implantation. ICRS macroscopic scoring was of cartilage regeneration presence or absence on bio-printed scaffold. Data are represented as means ± SD.

**Figure 4 cells-11-02934-f004:**
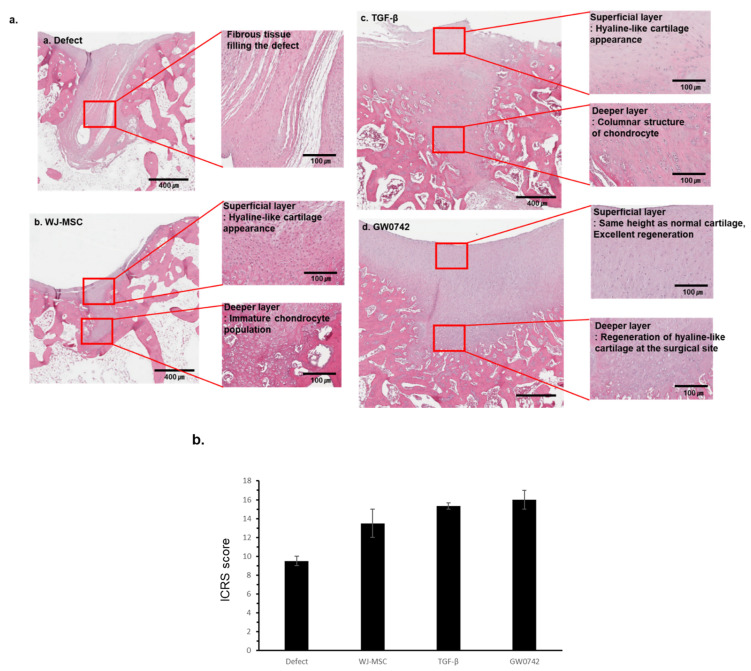
Histological analysis of repaired cartilage by bio-printed scaffold. Representative image of repaired cartilage at 12 weeks. (**a**) Histological observation of cartilage repair of a representative sample in each group at 12 weeks after implantation (a. defect, b. WJ-MSCs, c. TGF-β, d. GW0742; H&E staining); scale bar: 400 µm and 100 µm for each. (**b**) Histological evaluation of cartilage recovery using the ICRS Visual Histological Assessment Scale and group comparisons at 12 weeks. Data are represented as means ± SD.

**Figure 5 cells-11-02934-f005:**
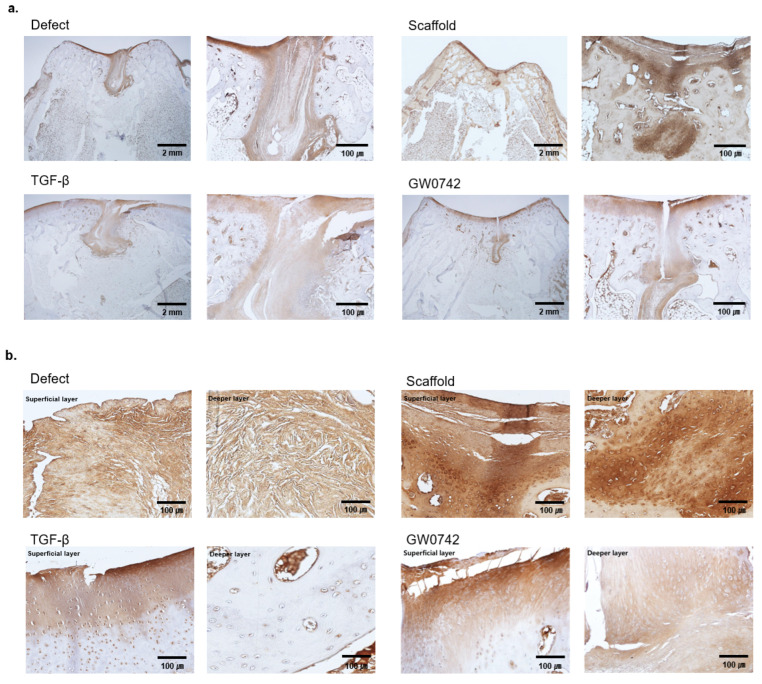
Immunohistology staining of collagen II. (**a**) Immunohistological observations of cartilage repair for a representative sample in each group after 12 weeks. Scale bar: 2 mm and 100 µm for each type II collagen staining. (**b**) Histological observations of each layer of the groups. The figure shows a representative sample in each group after 12 weeks. Scale bar: 100 µm; type II collagen staining.

**Table 1 cells-11-02934-t001:** Sequence of primers used for RT-PCR studies.

Gene	Forward Primer (5′-3′)	Reverse Primer (5′-3′)
COL2A1	GGCAATAGCAGGTTCACGTACA	CGATAACAGTCTTGCCCCACTT
GAPDH	TCAACGGATTTGGTCGTATTGGG	TGATTTTGGAGGGATCTCGC

## Data Availability

The data presented in this study are available on request from the corresponding author.

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
