# Peer review of "PPARδ Agonist Promotes Type II Cartilage Formation in a Rabbit Osteochondral Defect Model"

_cells, 2022, doi:10.3390/cells11192934_

Round 1
Reviewer 1 Report
The authors present a novel technique for cartilage restoration in a rabbit cartilage osteochondral defect model using a Peroxisome proliferator-activated receptor delta (PPARδ) agonist, GW0742, infused 3D collagen scaffold to induce type II cartilage from MSCs in this study.
The authors hypothesized that combining the PPARδ agonist, GW0742, with a 3D collagen scaffold would enhance chondrogenic differentiation in this study. Based on previous studies and their 3D printing, they used PPARδ-infused 3D collagen scaffolds to promote type II collagen formation in a rabbit osteochondral defect model.
The study, which started with a correct material method, was reinforced with a correct histological quantitative analysis.
As the authors say, although PPARβ/δ agonists may not be beneficial for clinical use to treat type II diabetes or dyslipidemia considering the risks when administered systemically, PPARβ/δ administration for musculoskeletal applications, such as promotion of osteogenesis for fracture healing and induction of MSC differentiation to chondrocytes to promote cartilage regeneration is appealing.
At the end of the study, the authors pointed out that consideration should be given to the fact that the effect of PPARδ agonists on MSCs in this study differ from the concerns mentioned above for several reasons. Those are;
First, the MSCs and PPARδ agonist GW0742 were contained in the collagen-based scaffold and did not produce systemic effects. Second, the shape and size of the scaffold were precisely tailored to fit the damaged cartilage and implanted into the damaged site, providing a secure, stable construct for integration into the host tissue and local control of MSC differentiation. Third, the cartilage microenvironment influenced the differentiation of MSCs into chondrocytes, which possess a low proliferation rate and metabolic activity. Lastly, PPARδ agonists possess immunomodulatory properties, such as anti-inflammatory properties, which reduce the activation of infiltrated macrophages and restore balance to the proinflammatory state seen in cartilage damage models so that the type II collagen can develop. The use of PPARδ with MSC embedded scaffolds in type II hyaline cartilage regeneration has been demonstrated as a promising method in a rabbit model without side effects.
The authors concluded that the future studies should evaluate the biomechanical properties of the regenerated cartilage and compare the efficacy of other PPARδ agonists that similar properties with the hope of achieving type II hyaline cartilage regeneration with native biomechanical properties for human use.
I liked the study very much, both methodologically and in terms of the results it achieved.
Author Response
The authors appreciate the reviewer’s comment, so there is no point in answering.
Reviewer 2 Report
Here the authors have developed a novel strategy to regenerate type II collagen using bio-printed scaffolds consisting of collagen and peroxisome proliferator-activated receptor delta, (PPARδ) agonist, GW0742. They demonstrate useful application of this technology using a rabbit cartilage osteochondral defect model to induce type II cartilage from MSCs.
While this is a novel application and an interesting study, at no point do the authors show any statistically significant differences across treatments despite mentioning statistical analysis in their Methods section. The authors make statements like "these results indicate that GW0742 is more potent than TGF-β in inducing cartilage regeneration" yet have no actual statistics to back up their claims.
In Figure 3, the legend states, "Data are represented as mean ± SD" which is not true as there are no visible error bars.
Author Response
Comment: While this is a novel application and an interesting study, at no point do the authors show any statistically significant differences across treatments despite mentioning statistical analysis in their Methods section. The authors make statements like "these results indicate that GW0742 is more potent than TGF-β in inducing cartilage regeneration" yet have no actual statistics to back up their claims.
In Figure 3, the legend states, "Data are represented as mean ± SD" which is not true as there are no visible error bars.
Answer: Authors found there missing points about statistical indications in this paper, and I added the statistical indications to highlight the potency of GW0742.
Furthermore, Authors added error bars in Figures 3 and 4.
Reviewer 3 Report
Manuscript Title: PPARδ agonist promotes Type II cartilage formation in a rabbit 2 osteochondral defect model
Manuscript ID: cells-1882804
Journal: cells
Decision : Accept with major revision
The authors have checked the chondrogenic capacity of peroxisome proliferator-activated receptor delta, (PPARδ) agonist GW0742, infused 3D collagen scaffold for cartilage restoration in a rabbit cartilage osteochondral defect model. They found induction of type II cartilage from MSCs. However, there are some points that are needed to be addressed before acceptance of this manuscript.
Comment 1. In vitro studies should be conducted for support of the in vivo results.
Comment 2. The bioprinted scaffolds should be checked for attachment of MSCs and differentiation into chondrogenic lineage in vitro.
Comment 3. The characterization data of the scaffolds should be presented like porosity, elasticity, and other associated mechanical properties that might play a vital role in the cartilage formation.
Comment 4. Expression levels of chondrocyte marker genes and/or proteins at different time intervals should be given to establish the mechanism and confirmation of the differentiation into chondrogenic lineage

Author Response
The authors appreciate the reviewer’s comment. We added new figures and responded according to the reviewer's statements.
Comment 1. In vitro studies should be conducted to support the in vivo results.
Answer: Previous studies containing in vitro results about PPARδ agonist (GW0742) promote chondrogenesis of MSCs in vitro and revealed the mechanism that the authors already mentioned in the paper. Please check the below references.
- Heck, B.E.; Park, J.J.; Makani, V.; Kim, E.C.; Kim, D.H. PPAR-δ Agonist With Mesenchymal Stem Cells Induces Type II Collagen-Producing Chondrocytes in Human Arthritic Synovial Fluid. Cell Transplantation. 2017, 26, 1405–1417.
- Kress, B.J.; Kim, D.H.; Mayo, J.R.; Farris, J.T.; Heck, B.; Sarver, J.G.; Andy, D.; Trendel, J.A.; Heck, B.E.; Erhardt, P.W. Synthesis and Evaluation of PPARδ Agonists That Promote Osteogenesis in a Human Mesenchymal Stem Cell Culture and in a Mouse Model of Human Osteoporosis. J Med Chem. 2021, 64, 6996-7032.
- Kim, D.H.; Kim, D.H.; Heck, B.E.; Shaffer, M.; Yoo, K.H.; Hur, J. PPAR-δ agonist affects adipo-chondrogenic differentiation of human mesenchymal stem cells through the expression of PPAR-γ. Regenerative Therapy. 2020, 15, 103-111
Comment 2. The bioprinted scaffolds should be checked for attachment of MSCs and differentiation into chondrogenic lineage in vitro.
Answer: Authors added figures of macroscopic and microscopic observation of thin scaffolds which contain MSCs and those results revealed that MSCs are attached to the scaffold during chondrogenesis until day 15. Moreover, added figures including alcian blue staining on day 15 support that MSCs are were intact and had differentiating capacity into chondrocytes.
Comment 3. The characterization data of the scaffolds should be presented like porosity, elasticity, and other associated mechanical properties that might play a vital role in the cartilage formation.
Answer: Unfortunately biomechanical properties could not be measured at this time. However, our observation is that mechanical properties like the degree of strength can be changed depending on the temperature. For example, at a temperature of around 25 degrees, the scaffold maintains mechanical intensity but it would be softer in the incubator (around 37 degrees). This is because the scaffold is mostly composed of collagen which its melting point is around 43.2 degrees. Reference (Irawan, V.; Sung, TC.; Higuchi, A.; Ikoma, T. Collagen Scaffolds in Cartilage Tissue Engineering and Relevant Approaches for Future Development. Tissue Eng Regen Med. 2018, 15, 673-697.) indicated that scaffold with collagen increase mechanical and bioactivities. In addition, reference (Ghodbane, SA.; Dunn, MG. Physical and mechanical properties of cross-linked type I collagen scaffolds derived from bovine, porcine, and ovine tendons. J Biomed Mater Res A. 2016, 104, 2685-2692.) showed collagen from any species can be suitable materials for various tissue engineering applications.
Comment 4. Expression levels of chondrocyte marker genes and/or proteins at different time intervals should be given to establish the mechanism and confirmation of the differentiation into chondrogenic lineage
Answer: Authors added a new figure reverse transcription-PCR result that human collagen type 2 expression level from the scaffold at different conditions (groups: scaffold containing WJ-MSC only and scaffold containing WJ-MSCs with TGF-β or GW0742) to support cells in the scaffold express chondrocyte marker at end of chondrogenesis. But we couldn’t evaluate chondrocyte markers at different time points to support the mechanism at this time.
Please kindly check reference no. 12 (Kim, D.H.; Kim, D.H.; Heck, B.E.; Shaffer, M.; Yoo, K.H.; Hur, J. PPAR-δ agonist affects adipo-chondrogenic differentiation of human mesenchymal stem cells through the expression of PPAR-γ. Regenerative Therapy. 2020, 15, 103-111.) already revealed that the mechanism of PPARδ induces the chondrogenesis of MSCs via inhibiting PPARγ expression which is related to anti-inflammatory and pro-adipogenic effects and induces type 2 collagen results in enhancing chondrogenesis of MSCs
Round 2
Reviewer 3 Report
Authors have responded satisfactorily and hence the manuscript can be accepted.